# Developmental Ultrasound Characteristics in Guinea Pigs: Similarities with Human Pregnancy

**DOI:** 10.3390/vetsci10020144

**Published:** 2023-02-10

**Authors:** Alejandro A. Candia, Tamara Jiménez, Álvaro Navarrete, Felipe Beñaldo, Pablo Silva, Claudio García-Herrera, Amanda N. Sferruzzi-Perri, Bernardo J. Krause, Alejandro González-Candia, Emilio A. Herrera

**Affiliations:** 1Laboratorio de Función y Reactividad Vascular, Programa de Fisiopatología, Instituto de Ciencias Biomédicas (ICBM), Facultad de Medicina, Universidad de Chile, Santiago 7500922, Chile; 2Institute of Health Sciences, University of O’Higgins, Rancagua 2841959, Chile; 3Departamento de Ingeniería Mecánica, Facultad de Ingeniería, Universidad de Santiago de Chile, Santiago 9170022, Chile; 4Department of Physiology, Development and Neuroscience, University of Cambridge, Cambridge CB2 3EG, UK; 5International Center for Andean Studies (INCAS), University of Chile, Putre 1070000, Chile

**Keywords:** gestation, Doppler ultrasound, fetal growth, placental function, animal model

## Abstract

**Simple Summary:**

Despite the relevance of biometrical and blood flow assessments for studying fetoplacental physiology during pregnancy, there is no detailed description of any animal model, which is needed to extrapolate results to human pregnancy. Here, we examined biometry and intrauterine blood flow in pregnant guinea pigs from the second trimester until term. We show that fetal and placental biometry, as well as changes in the main vascular beds across pregnancy, compared qualitatively to data from humans. These findings emphasize that the guinea pig is a reliable model to study fetal development and placental function with translational significance for human pregnancy.

**Abstract:**

Background: Biometrical and blood flow examinations are fundamental for assessing fetoplacental development during pregnancy. Guinea pigs have been proposed as a good model to study fetal development and related gestational complications; however, longitudinal growth and blood flow changes in utero have not been properly described. This study aimed to describe fetal and placental growth and blood flow of the main intrauterine vascular beds across normal guinea pig pregnancy and to discuss the relevance of this data for human pregnancy. Methods: Pregnant guinea pigs were studied from day 25 of pregnancy until term (day ~70) by ultrasound and Doppler assessment. The results were compared to human data from the literature. Results: Measurements of biparietal diameter (BPD), cranial circumference (CC), abdominal circumference, and placental biometry, as well as pulsatility index determination of umbilical artery, middle cerebral artery (MCA), and cerebroplacental ratio (CPR), were feasible to determine across pregnancy, and they could be adjusted to linear or nonlinear functions. In addition, several of these parameters showed a high correlation coefficient and could be used to assess gestational age in guinea pigs. We further compared these data to ultrasound variables from human pregnancy with high similarities. Conclusions: BPD and CC are the most reliable measurements to assess fetal growth in guinea pigs. Furthermore, this is the first report in which the MCA pulsatility index and CPR are described across guinea pig gestation. The guinea pig is a valuable model to assess fetal growth and blood flow distribution, variables that are comparable with human pregnancy.

## 1. Introduction

Biometrical and blood flow assessments are currently the best tools for examining in utero development, cardiovascular health, and the diagnosis and management of several high-risk pregnancy conditions, such as preeclampsia, congenital diseases, and placental insufficiency [1,2,3]. In terms of blood flow measurements, the most explored vascular beds are the uterine (UtA), umbilical (UA), and middle cerebral artery (MCA). Of note, abnormalities in UtA blood flow by Doppler ultrasound are associated with hypertensive disorders, and they are a good predictor of neonatal adverse outcomes [4], whereas changes in UA and MCA are useful in the diagnosis and management of fetal growth restriction (FGR) [3]. In addition, the cerebroplacental ratio (CPR), defined as the relationship between the pulsatility indices (PIs) of the MCA and UA, has been proposed as a helpful tool to forecast adverse fetal outcomes, such as perinatal death in pregnancies with suspected FGR [5]. Furthermore, abnormal CPR during the second trimester is associated with an increased risk of preterm delivery and neonatal intensive care unit admission [6]. There is also a potential benefit of identifying blood flow changes in utero in stratifying babies for postnatal monitoring and potential risks. For instance, findings suggest that abnormal CPR may predict delayed neurodevelopment in early childhood born with FGR [7]. Despite these data, more studies are necessary to determine its clinical utility and to identify experimental models to investigate the pathophysiological changes impacting prenatal and postnatal outcomes.

Several experimental animal species have been employed to study fetal development and placental function, with specific attention paid to examining placental histological, gene, and protein expression changes that are relevant to maternal and fetal health [8,9,10,11]. However, there is no well-established animal model which allows the association of in-vivo vascular flow changes with such histological and molecular changes during pregnancy, which is needed to provide an avenue for translating findings to humans.

Guinea pigs have been proposed as an excellent model to study placental function, fetal development, and intrauterine programming [8]. This is based on the guinea pig having a hemomonochorial placenta, similar to humans [12], as well as a deeper trophoblast invasion of the decidua compared to rodents [13]. They are also amendable to instrumentation, such as for the insertion of catheters and vascular occluders, as well as to improve the investigation of blood flow changes during pregnancy [14,15]. Most studies employing guinea pigs have focused on measuring placental structure, nutrient exchange, and offspring outcomes [8]. Few studies have used ultrasound to describe biparietal diameter (BPD) and the resistance index of the umbilical artery during guinea pig pregnancy (G22 to G69) [16]. Further studies have analyzed other biometric parameters, such as abdominal and placental diameter with ultrasound imaging in guinea pigs [17,18]. Lastly, studies using sonography have reported that UA and UtA PI are altered in guinea pig pregnancies with placental insufficiency [15], with changes similar to what is seen in humans. Despite these reports, no study has assessed fetal and placental biometry in association with changes in blood flow characteristics, particularly regarding MCA and CPR using Doppler ultrasound across pregnancy, in guinea pigs.

Therefore, the aims of this study were to describe the normal fetal and placental growth, as well as the Doppler assessment of several arteries across pregnancy in guinea pigs, and to discuss the similarities of these data with those obtained for human pregnancy.

## 2. Materials and Methods

All animal care, maintenance, and procedures were approved by the Institutional Animal Care and Use Committee (CICUA) of the Universidad de Chile (20354-MED-UCH) and performed under the agreement to Guidelines for the Care and Use of Laboratory Animals. Ultrasonographic examinations were performed in conscious, non-sedated animals, gently restrained by an expert operator.

### 2.1. Animals

Nineteen adult female Pirbright White guinea pigs (Cavia porcellus) were used in this study. Animals were housed under standard conditions (35–40% humidity, 20–21 °C, and 12 h/12 h light/dark cycle) and fed standard chow that was controlled according to body weight (LabDiet 5025, Guinea Pigs, 25–30 g/day). At 3–4 months of age, virgin sows in estrus, determined by the observation of vaginal membrane opening, were paired with a male for 2 days. After this mating period, the females returned to individual housing cages with daily monitoring of body weight, food intake, and water consumption. At gestational day (G) 20–25, pregnancy was confirmed by ultrasonography, considering the first day paired with the male as G0 [15]. After confirming pregnancy, the guinea pigs were submitted twice a week to ultrasound and Doppler examination until G35 (n = 4 dams/10 fetuses), G60 (n = 4 dams/11 fetuses), or birth (G70 ± 3, n = 11 dams/23 fetuses) after being randomly assigned to each group. Animals euthanized at G35 and G60 were submitted to a maternal anesthetic overdose (sodium thiopentone 200 mg·kg^−1^ i.p., Opet, Laboratorio, Chile). After the cardiorespiratory arrest was confirmed, fetuses and placentas were dissected, and we measured biparietal diameter (BPD), fetal weight and volume, and placental weight, thickness, length width, and volume. Lastly, placental efficiency was calculated as fetal weight/placental weight.

### 2.2. Ultrasound Assessment

Biometrical and vascular assessments were performed using a portable ultrasonograph (Z6 VET, Mindray) with an L14-6P linear ultrasound transducer. BPD, cranial circumference (CC), abdominal circumference (AC), and placental thickness, transversal axis length, transversal area, and discoid perimeter and area were measured twice per week from G25 to the end of gestation. Placental volume was estimated as follows:Placental volume (mm^3^) = (discoid placental area) × (placental thickness).

Blood flow velocity in the umbilical artery (UA), middle cerebral artery (MCA), and both uterine arteries (UtA) were assessed by Doppler ultrasound, and the pulsatility index (PI) was estimated by the ultrasonography software. UtA was determined as the average of the left and right arteries during each examination. The cerebroplacental ratio was calculated as MCA PI to UA PI. The definition of each measurement is shown in Table 1. Ultrasound accuracy was established by calculating the percentage error between the ultrasound and euthanized measurements as follows: [(Postmortem direct measurement − in vivo ultrasonic measurement)/postmortem direct measurement] × 100.

### 2.3. Ultrasound Data Processing

For the Doppler ultrasound analyses (PI UA, PI MCA, and CPR), the best fit of the curve was determined and plotted against percentage of pregnancy (PP) (2.5% of PP between points). Then, we calculated the initial slope as the first data points before reaching a 5% of variation in the best-fit slope between one data point and the next. This slope was designated as velocity 1 (V1). In a similar way, a final slope was calculated including the data points after the percentage variation in the best-fit slope returned lower than 5%. This was designated as velocity 3 (V3). In addition, when possible, an inflection point was determined, which was defined as the point where the slope was equal to 0. Lastly, for Doppler data, the percentage variation between the initial and final slopes was calculated.

Human data related to biometry from a multinational prospective observational study [19] and Doppler ultrasound data summarized in a recent systematic analysis [20] were reviewed. The PP was calculated considering 40 weeks and 70 days as 100% pregnancy in humans and guinea pigs, respectively. For the biometry data (BPD, CC, and CA), we estimated the percentage of growth (PG) of each variable, considering 100% as the value at 40 weeks and 70 days of gestation in humans and guinea pigs, respectively. We plotted the biometrical and Doppler data against PP for each variable.

Estimated fetal weight (EFW) was calculated from BPD, AC, and the measured weight at four timepoints: G30 and G60 (euthanized), and G69 and G72 (post delivery). Using data obtained at those timepoints, we determined a multiple linear regression between the decimal logarithm of the fetal/neonatal weight and the explanatory variables, including the interaction between them, obtaining a Hadlock-type equation to determine the EFW [21,22].

### 2.4. Histology

The collected placentas at G35 and G60 were immersed in 4% formaldehyde for 24 h and then stored in PBS until analysis. Briefly, the samples (half of placenta) were paraffin-embedded with the longest axis oriented to the front, and 5 µm tissue slices were obtained and stained with hematoxylin/eosin using a standard protocol [23]. Digital images of each stained section were acquired on a Nanozoom Slide Scanner (Hamamatsu, Japan) and viewed with NPD2.view 2 software. 

### 2.5. Statistics

Guinea pig biometrical and Doppler data were plotted against the gestational day. Curves were adjusted according to their best fit, which was determined by the sum-of-squares F-test. Biometrical variables at the end of gestation were expressed as the mean ± SEM. Statistical comparisons between ultrasound and euthanized measurements were made by a paired *t*-test (significant if *p* < 0.05) using Graph Pad Prism 9.0 (San Diego, CA, USA).

## 3. Results

### 3.1. Fetal Biometry

The BPD, CC, and AC of fetuses were measured using ultrasound in guinea pigs from G25 until term. This analysis revealed that BPD, CC (perimeter), and AC (perimeter) increased along with gestational age and conformed to a second-order polynomial (quadratic) curve, all with high determination coefficients (Figure 1A,B,D). Fetal CC (area) and AC (area) also increased with gestational age; however, these parameters fitted to a first-order polynomial (straight line) curve (Figure 1C,E).

### 3.2. Placental Biometry

Placental thickness, placental length, and transversal area of the placenta were visualized from G25 until term. These parameters were found to increase with gestational age and fitted to a quadratic function (Figure 2A–C). The perimeter and area of discoid placental view were also fitted to a quadratic line with a low determination coefficient. Placental discoid area was difficult to quantify in ultrasound assessments performed after G50, as the placental diameter was larger than the maximum dimension of the screen (Figure 2D,E). Lastly, placental volume was calculated, and the best fit was obtained when placental weight was fitted to a straight line (Figure 2F).

### 3.3. Doppler Ultrasound

In guinea pigs from G25 until term, UA PI was examined and described as a quadratic negative nonlinear regression (Figure 3A), whereas MCA PI was detected from G35 until G65 and fitted to linear regression (Figure 3B). Both PI measures decreased across pregnancy. We found that CPR fitted to a quadratic nonlinear regression but with a high variability (r^2^ = 0.2030) (Figure 3C). Lastly, UtA PI showed a straight-line fit with a high variability across gestation (r^2^ = 0.1004) (Figure 3D).

### 3.4. Fetal and Placental Biometry

Biometrical data measured at euthanasia are shown in Table 2. By comparison (Wilcoxon matched-pairs signed rank test) to values obtained with ultrasound measurements at G35, no significant differences were identified, with an estimated ultrasound accuracy of 1.17%. Using histological analyses, three macrostructures of the placenta could be identified at G35: the decidua, subplacenta, and labyrinth + interlobium (Figure 4A). However, at G60 the subplacenta was absent (Figure 4B). Ultrasound screening was unable to differentiate placental structures at any gestational age studied.

### 3.5. Fetal Biometry at Birth

We assessed weight and BPD at birth in both females and males. Although no significant differences in weight and BPD between sexes were identified (Figure 5A,B), the ratio of BPD to body weight was greater in males compared to females (Figure 5C).

### 3.6. Estimated Fetal Weight

After performing a multiple linear regression considering the variables BPD, AC, and their interaction, as well as the fetal/neonatal weight, at the four different timepoints, we obtained the Hadlock-type equation for estimated fetal weight (Figure 6).

### 3.7. Comparison of Guinea Pig and Human Fetal Biometry

The growth rates of BPD, CC, and CA as a percentage of pregnancy for our guinea pig data and humans (published data from WHO 2017 reports [19]) are plotted in Figure 7A–C. Qualitatively, the three parameters displayed similar growth patterns as pregnancy progressed when comparing guinea pigs to humans (Figure 7A–C).

### 3.8. Guinea Pig and Human Doppler Ultrasound

Comparison of PI of UA, PI of MCA, and CPR as a percentage of pregnancy for our guinea pig data and human published studies (Medina-Castro et al. [24,25], Parra-Cordero et al. [26], Arduini et al. [27], Seffah et al. [28], Bahlmann et al. [29], Morales-Roselló et al. [30], Ebbing et al. [31], and Baschat et al. [32], previously summarized in Oros et al. 2019 [20]) are shown in Figure 7D–F. This analysis revealed that, in both guinea pig and human datasets, UA PI decreased across pregnancy, with a tendency to stabilize in the last third of pregnancy (Figure 7D). Meanwhile, MCA PI in humans exhibited a hyperbola-like nonlinear regression, reaching its maximum point (inflection point) at around 70% pregnancy. However, in guinea pigs, MCA PI showed a nonlinear regression, decreasing across pregnancy with a change in velocity at 77% pregnancy progression and a 76% variation throughout pregnancy (Figure 7E). In the case of CPR, this conformed to a similar hyperbola-like nonlinear regression in both human and guinea pig datasets, reaching a maximum point at 80% and 65% of pregnancy in humans and guinea pigs, respectively (Figure 7F).

## 4. Discussion

In this work, we described fetal and placental growth, as well as intrauterine hemodynamic changes that occur during physiological, healthy guinea pig pregnancies, and then compared these data to those available for humans. This is important for understanding the usefulness and translatability of fetoplacental physiological studies in guinea pigs. Furthermore, to the best of our knowledge, this is the first report of MCA PI and CPR across pregnancy in guinea pigs.

BPD and AC are the main parameters assessed to determine fetal growth in humans and other species [3,33,34]. In the current study, we showed that BPD, CC, and AC are reliable variables when assessing fetal growth in guinea pigs. Here, we decided to take a data-driven approach to model our results. This was based on the fact that our data fitted best to a straight or quadratic model rather than to classic growth equations such as the exponential growth equation (Y = Y0 × exp(k × X)) or the logistic growth model (Y = YM × Y0/((YM − Y0) × exp(−k × x) + Y0), when comparing by sum-of-square F-tests.

Previously, Turner and Trudinger (2000) [35], and Santos et al. (2014) [17] described the normal growth of BPD in guinea pigs from G22 until the end of pregnancy. Turner and Trudinger presented these data as a mean of 3–5 day intervals, whereas Santos et al. showed a mean for every 5 days of gestation. Interestingly, they reported mean values of around 1.9 cm and 3.4 cm, respectively, at G65. In our work, the mean value at G65 was 1.86 cm, which is very similar to that reported by Turner and Trudinger. Differences in values between studies likely relate to factors such as the breed/strain or maternal age of guinea pigs used. For instance, Turner and Trudinger reported a breed which was different to ours (English Short Hair vs. Pirbright White, respectively), as well as a higher range of maternal age (4–12 months vs. 3–5 month) [35]. Differences could also be related to the operator’s technique and variability in the litter size of the guinea pigs in each study [36]. In a similar way, we found similar values in fetal weight at G35 and G60 when we compared our postmortem results to Draper et al. [37] and Ibsen et al. [38]. For instance, at G35, the differences in fetal weight between our results and Draper et al. were −1.54 g, with an average of 3.06 g and 4.6 g, respectively. Again, these small differences could be related to the guinea pig strain used, which was not reported in those studies, as well as to the litter size employed, which varied from one to eight fetuses. Our study used dams with litter sizes of two and three fetuses. Although this might be considered a limitation, we provided consistent fitting of variables such as the BPD and pulsatility index of umbilical artery in virgin dams of 606.9 ± 94.5 g. Moreover, by studying dams with only two or three fetuses, we limited the effect of varied litter size as a confounder in our analyses.

In the current investigation, biometry at birth (BPD and CC) was similar between females and males; however, symmetry of neonates appeared to differ between the sexes, with males showing a greater ratio of BPD to body weight. This aspect could be relevant to explore when studying fetal development. Unfortunately, we were unable to diagnose fetal sex by ultrasound during pregnancy; therefore, we were unable to compare intrauterine changes between males and females.

Studies have previously proposed that placental shape and size are relevant for identifying FGR as well as the programming of offspring cardiovascular risk [39,40]. With this in mind, we attempted to describe placental growth across pregnancy by taking measures of placenta size using two different axes. The transversal view allowed us to capture a longitudinal and transverse axis, where we visualized the decidua and the exchange placental region. This observation and histological studies of placentas at G35 and G60 are consistent with Santos et al. [17], likely reflecting changes in lobule formation and vascular organization of the guinea pig placenta toward term. Moreover, histological inspection of the placenta confirmed the involution of the subplacenta across pregnancy as it was no longer apparent at G60 [41]. Further work is required to develop strategies to differentiate this structure, along with other placental compartments, by ultrasound, as well as to develop quantitative analysis of the postmortem placental structure. Further work is also needed to examine placental growth from a discoid view as, after G50, it was difficult to visualize it in only one screenshot. Indeed, imaging the placenta in a discoid view could further inform on the placental shape and vascular flow in guinea pigs [40]. Other imaging techniques may be useful to assessing placental size and morphological characteristics, such as microCT imaging and CT angiography [42,43].

We assessed UA and MCA PI during guinea pig gestation, as these are probably the two most used Doppler measurements to monitor fetal wellbeing and FGR during pregnancy [3]. Here, we showed that UA PI exhibited an inverse nonlinear regression, with a good fit during pregnancy in guinea pigs. On the other hand, MCA PI described an inverse linear regression but with a moderate determination coefficient and a higher variability. Taking these indices together, we were able to estimate the CPR; however, our data were again highly variable. Furthermore, visualization of MCA was feasible only after G35, which may provide an explanation for the linearity of the data fit and the high variability observed mostly at earlier gestational age.

In humans, an absent end-diastolic flow in the umbilical artery or changes in UA PI centile (i.e., >95th centile) associated with structural changes (i.e., <10th centile in estimated fetal weight) are used for the diagnosis of early- and late-onset FGR, respectively [3]. Thus, in humans, an alteration of PI is determined by changes in PI centile. However, in guinea pigs, an alteration or significant change in PI is determined by statistical comparison between two experimental groups. Previously, our group reported that the umbilical artery PI is increased in guinea pigs with placental insufficiency [15]. In the current study, we provide new information regarding the behavior of both functional (Doppler) and structural (biometrical) variables in a physiological context. However, further work in our lab is currently underway to study the role of possible alterations in these parameters under different context (e.g., gestational hypoxia).

EFW is a valuable parameter in the clinical context [3]. In humans, it is calculated on the basis of at least two biometrical parameters, such as head circumference and abdominal circumference [22]. Considering the differences in head shape between humans and guinea pigs, we decided to replace head circumference with BPD, since the latter seems to be more consistently measured. From there, we determined a Hadlock-type equation for EFW. However, the utility of this equation for our study is limited. Firstly, our study involved data obtained from four timepoints. Secondly, the number of litters studied was limited. Lastly, we were unable to validate our findings against a different cohort of guinea pigs. Despite this, we think that this equation may be helpful for other researchers working with guinea pigs. Indeed, current efforts in the lab are aimed at validating this parameter so as to increase the knowledge of guinea pig prenatal growth.

Applying a more in-depth analysis, we tried to establish similarities in the behavior of biometrical and blood flow data measured by ultrasound across pregnancies in guinea pigs and humans. Among the studied biometrical variables, fetal BPD and CC seemed to have a similar growth pattern between guinea pigs and humans, whereby these parameters increased with gestation. On the other hand, UA PI tended to decrease along gestation in guinea pigs and humans. In addition, the linear reduction in MCA PI during guinea pig gestation differed from the nonlinear behavior of human data. One possible explanation for these differences could be related to the gestational age at which the variables were initially assessed, as we obtained data at 40% pregnancy in guinea pigs, whereas most human data are reported at 60% pregnancy. Lastly, values for CPR showed a similar behavior in humans and guinea pigs. In humans, while there is high variability among different studies, and while the reference charts for Doppler parameters have yet to be fully established [20,44], our results in guinea pigs are promising as we were able to report similar clinically relevant values, e.g., for UA PI and CPR.

## 5. Conclusions

Altogether, our findings increase the knowledge on fetal and placental growth, as well as hemodynamic assessment across pregnancy in guinea pigs. This is important as guinea pigs are established as a valuable animal model for studying pregnancy complications and the developmental programming of heath and disease [8]. Furthermore, the docile nature, ease of reproduction and handling, and body size of guinea pigs make them ideal for studies of development and reproduction. This study showed that fetal and placental biometry, as well as changes in the main vascular beds across pregnancy, can be comparable to data from humans. These findings emphasize that the guinea pig is a reliable model to study fetal development and placental function with translational significance for human pregnancy.

## Figures and Tables

**Figure 1 vetsci-10-00144-f001:**
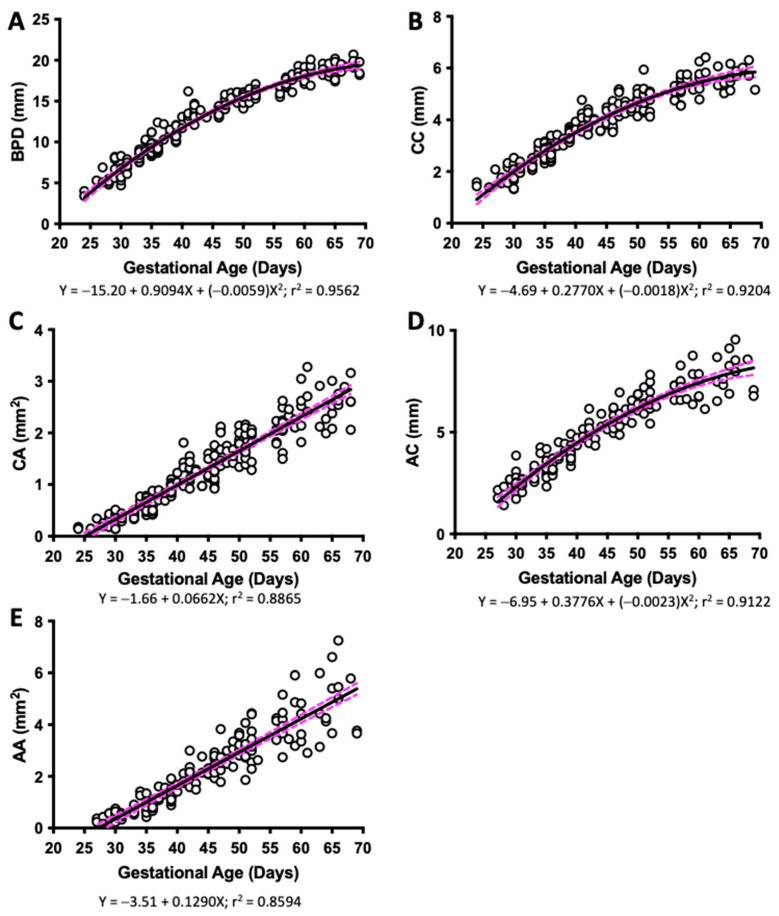
**Fetal biometry across pregnancy.** Guinea pig fetal (**A**) biparietal diameter (BPD), (**B**) cranial circumference (CC), (**C**) cranial area (CA), (**D**) abdominal circumference (AC), and (**E**) abdominal area (AA), and their best fit from gestational day 25 to the end of pregnancy. Each dot represents an individual measurement (n = 47 from 19 litters). Black line = data best fit. Magenta dashed lines = 95% CI.

**Figure 2 vetsci-10-00144-f002:**
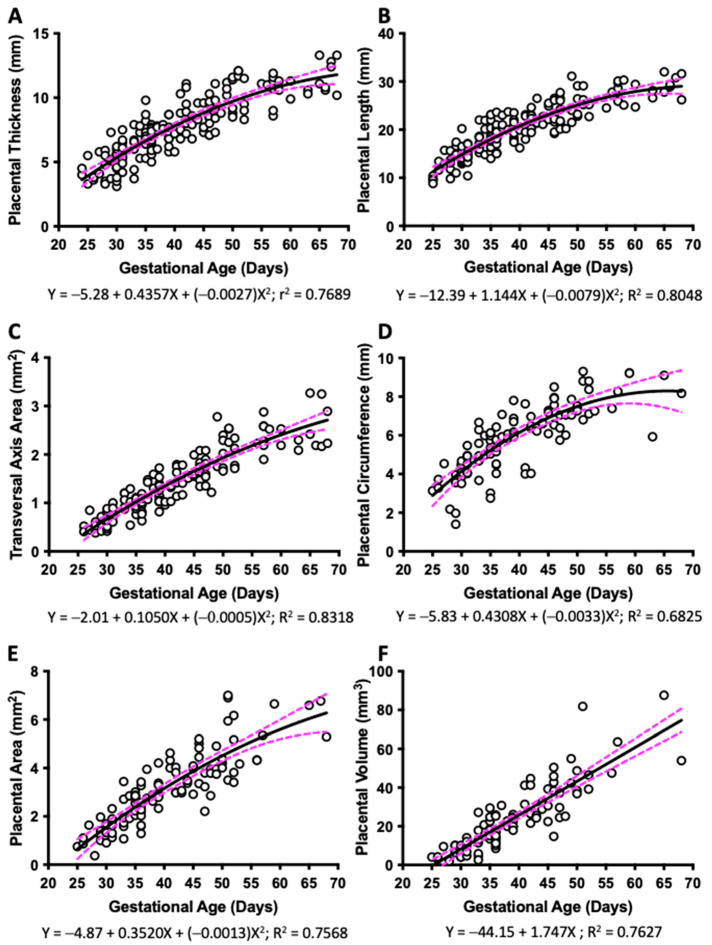
**Placental biometry across pregnancy.** Guinea pig placental (**A**) thickness, (**B**) length, (**C**) transversal area axis, (**D**) discoid perimeter, (**E**) discoid area, and (**F**) estimated placental volume, and their best fit from gestational day 25 to the end of pregnancy. Each dot represents an individual measurement (n = 47 from 19 litters). Black line = data best fit. Magenta dashed lines = 95% CI.

**Figure 3 vetsci-10-00144-f003:**
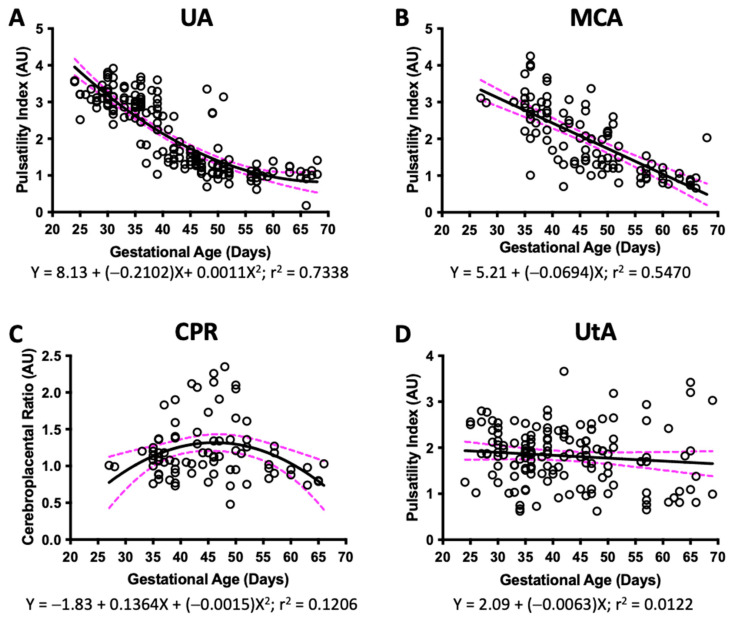
**Doppler ultrasound across pregnancy.** Guinea pig pulsatility index of (**A**) umbilical artery (UA), (**B**) middle cerebral artery (MCA), (**C**) cerebroplacental ratio (CPR), and (**D**) uterine artery (UtA) pulsatility index, and their best fit from gestational day 25 to the end of pregnancy. Each dot represents an individual measurement (n = 47 from 19 litters). Black line = data best fit. Magenta dashed lines = 95%CI.

**Figure 4 vetsci-10-00144-f004:**
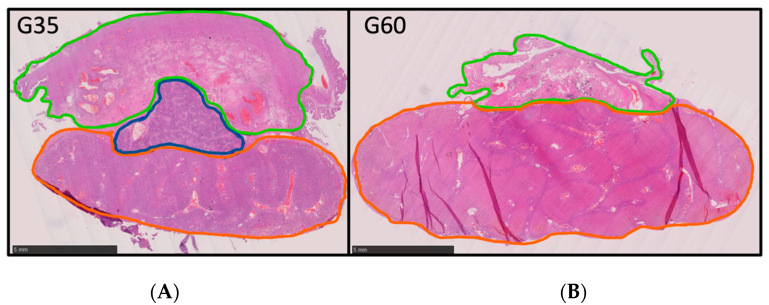
**Placental structure at G35 and G60.** Placental structure at (**A**) gestational day 35 and (**B**) gestational day 60. Green line = decidua. Dark-blue line = subplacenta. Orange line = interlobium and labyrinth zone. Scale = 5 mm.

**Figure 5 vetsci-10-00144-f005:**
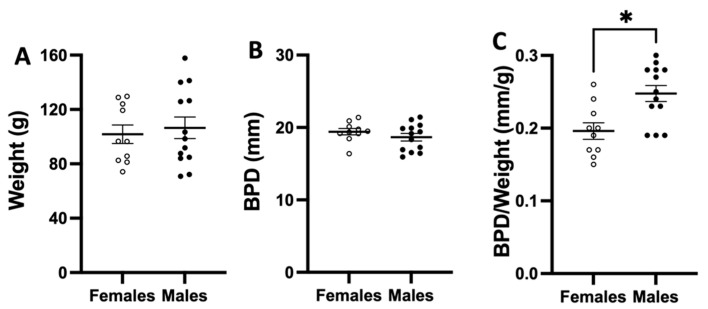
**Neonatal biometry at birth.** Neonatal (**A**) weight, (**B**) biparietal diameter (BPD), and (**C**) BPD to neonatal weight ratio of female (white circles, n = 10) and male (black circles, n = 13) guinea pigs. Data were analyzed by Mann–Whitney *t*-test. Significant differences when *p* < 0.05 (*).

**Figure 6 vetsci-10-00144-f006:**
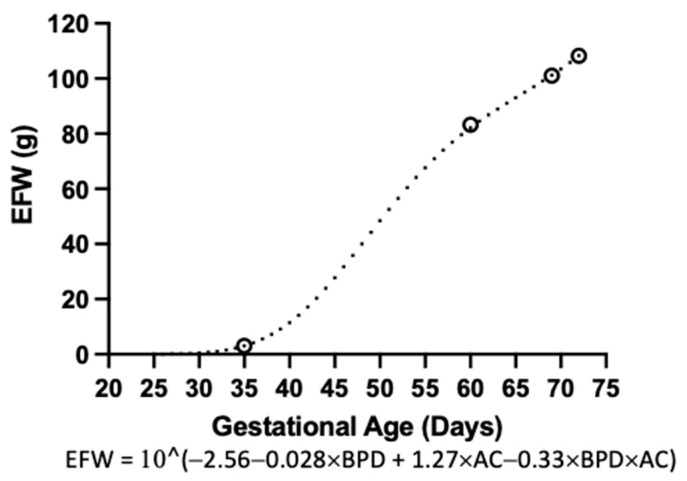
**Guinea pig estimated fetal weight.** Estimated fetal weight multiple linear regression from gestational day 25 to 72. White circles indicate timepoints where the variables were obtained (G35, n = 10; G60, n = 11; G69, n = 9; G72, n = 6). Black dots indicate the estimated fetal weight for gestational day.

**Figure 7 vetsci-10-00144-f007:**
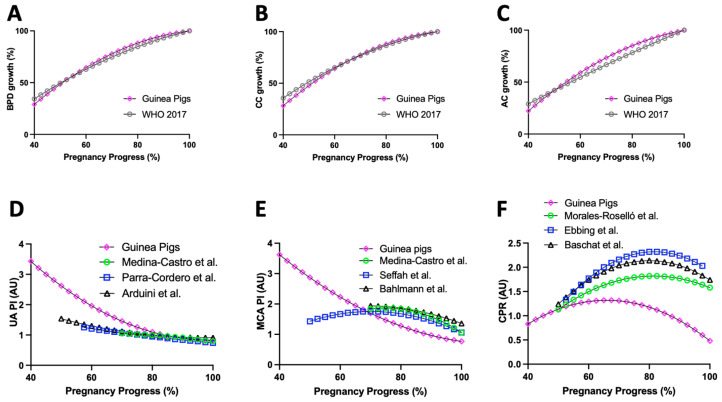
**Guinea pig and human fetal ultrasound variables across pregnancy.** Guinea pig and human (**A**) biparietal diameter (BPD), (**B**) cranial circumference (CC), and (**C**) abdominal circumference (AC) acceleration across pregnancy and pulsatility index (PI) of (**D**) umbilical artery (UA), (**E**) middle cerebral artery (MCA), and (**F**) cerebroplacental ratio (CPR) from 40% of pregnancy until term. WHO 2017 [19], Medina-Castro et al. [24,25], Parra-Cordero et al. [26], Arduini et al. [27], Seffah et al. [28], Bahlmann et al. [29], Morales-Roselló et al. [30], Ebbing et al. [31], and Baschat et al. [32] are studies based on human data.

**Table 1 vetsci-10-00144-t001:** Description of the ultrasound measurements made.

Variable	Definition	Units	Representative Image
Biparietal diameter (BPD)	From a transversal visualization of the cranium: Distance from the external side of the parietal bone to the internal side of the opposite parietal bone	mm	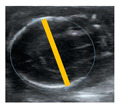
Cranial circumference (CC)	Elipse around DBP as transversal diameter and a longitudinal diameter touching the occipital bone	mm (perimeter), mm^2^ (area)	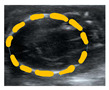
Abdominal circumference (AC)	Circumference around the edges of a transversal abdominal visualization at stomach/liver level	mm (perimeter), mm^2^ (area)	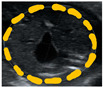
Transversal area of placenta (PA)	Region observed in a transversal visualization of placenta excluding decidua	mm (perimeter), mm^2^ (area)	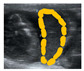
Placental length (PL)	Longitudinal segment in a transversal visualization of placenta	mm	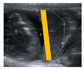
Placental thickness (PT)	Transversal segment in a transversal visualization of placenta	mm	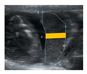
Discoid area of placenta	Region observed in a posterior visualization of placenta	mm (perimeter), mm^2^ (area)	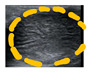
Umbilical artery (UA)	From a longitudinal view of the placenta: Blood flow detected by Doppler ultrasound between the fetus abdomen and the placenta	Pulsatility index ((systolic velocity-diastolic velocity)/mean velocity)	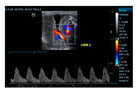
Middle cerebral artery (MCA)	From a transversal visualization of the cranium: Blood flow detected by Doppler ultrasound above the circle of Willis in a transverse direction	Pulsatility index ((systolic velocity-diastolic velocity)/mean velocity)	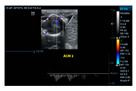
Uterine Artery (UtA)	From a sagittal view of the maternal urinary bladder: Blood flow detected by Doppler ultrasound surrounding the bladder in an ~45° angle	Pulsatility index ((systolic velocity-diastolic velocity)/mean velocity)	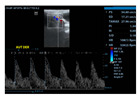
Cerebro-placental ratio (CPR)	Ratio between MCA and UA pulsatility indexes	AU	

**Table 2 vetsci-10-00144-t002:** Fetal and placental biometry at euthanasia.

	G35	G60
Fetal Weight (g)	3.06 ± 0.28	83.30 ± 3.47
BPD (mm) ^1^	9.21 ± 0.35	10.09 ± 0.35
Volume (mm^3^)	3.23 ± 0.16	79.00 ± 2.99
Placental weight (g)	2.51 ± 0.21	5.60 ± 0.23
Placental Thickness (mm)	6.12 ± 0.42	8.90 ± 0.71
Placental Length (mm)	19.43 ± 0.42	29.35 ± 0.74
Placental Width (mm)	14.43 ± 0.45	25.70 ± 0.57
Placental Volume (mm^3^)	2.74 ± 0.12	6.75 ± 0.39
Placental Efficiency (AU)	1.24 ± 0.08	15.06 ± 0.75

^1^ BPD, biparietal diameter.

## Data Availability

Not applicable.

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
