# Peer review of "Developmental Ultrasound Characteristics in Guinea Pigs: Similarities with Human Pregnancy"

_vetsci, 2023, doi:10.3390/vetsci10020144_

Round 1
Reviewer 1 Report
This is an ambitious undertaking, well performed overall, and makes a valuable contribution to the literature. It lacks in my view a historical perspective given that guinea pigs were common in laboratories prior to the ascendance of the mouse. I would have expected a nod to the highly cited study of prenatal growth in the guinea pig by Draper (Anat Rec 1920; 18 (4): 369-392) and perhaps to the wider study by Ibsen (J Exp Zool 1928; 51 (1): 51-93). Both papers can be downloaded from the respective journal’s web site.
An unreliable method was used to date pregnancy and this needs to be discussed in the MS. Many authors have taken advantage of the fact that guinea pigs mate postpartum, but that option is not available for the virgin guinea pigs used here. An alternative approach, which we have found reliable, is to observe the opening and closing of the vaginal membrane as described by Elvidge (J Inst Anim Technicians 1972; 23:111-117). Further accuracy can be obtained by taking vaginal smears (e.g. Han et al. Placenta 1999; 20: 361-77).
It is unsurprising to this reviewer that there was difficulty in visualizing the ductus venosus. Carter et al. (Lab Anim Sci 1992; 42: 174-9) demonstrated the absence of the DV in fetal guinea pigs confirming the situation observed at birth by Harman & Herbertson (Trans Kansas Acad Sci 1938; 41: 369-76). Their finding was in accordance with 98% of microspheres injected in the umbilical vein being recovered from the liver, which would not be expected with a patent DV. Thus, there is both morphological and functional evidence against DV flow in the guinea pig. The authors say this is the first time a DV has been identified (first para Discussion). If the authors stand by their claim, it will need to be discussed.
Data is presented for fetal and placental weight at G35 and G60. How does this compare with the data presented by Draper and Ibsen? Do the growth trajectories presented in Figures 1 and 2 differ from or agree with those drawn by these authors?
Guinea pigs are relevant because of their long gestation and precocial young. These characteristics are shared by other hystricomorph rodents. Recently advanced imaging techniques have been applied to the chinchilla (Mikkelsen et al. Roy Soc Open Sci 2017; 4: 161098) and the authors might consider whether that work is relevant.
Reviewer 2 Report
Overall this is a really interesting (although niche) manuscript with some interesting data. I have some comments mainly about the models used and the relationships between them.
Methods
How animals were assigned to each group should be stated.
UtA PI acquisition is not described in the methods (is this bilateral/unilateral)?
Do fetal size/US parameters depend on little size or maternal weight?
Possibly Fig1/2 could be redrawn with litter-mates marked?
Results
Relationships in Fig 1. The plotting is fine and data driven, but the implicit assumption in the model is that C is proportional to CC^2 (actually there is a mistake in Fig 1 C/E labels - change the notation for this and AC to areas.). This means that the straight line in Fig 1C is actually the locally straight part of a quartic equation (you can see this is roughly true by squaring the equation given in Fig 1B and plotting the result). This needs justifying somehow in the text as it is contradictory.
I don't disagree with the fitting that has been done, but the straight/quadratic equations are likely to themselves be approximations of more general growth models (based on exponentials). This should be discussed, if more data were to become available it would be possible to extend the models more accurately outside of the presented gestational age window. I would also be interested to see a plot of estimated fetal volume in Fig 2.
For Fig 2 - the same applies - please fix the axis label for Fig 2E, but in this case the link between placental length and transversal area is probably less simple than for CC because the placenta is probably less circular in cross section. This is probably the reason for the Placental Circumferential Area being curved rather than matching Fig1E.
I would also be interested to see a plot of estimated placental volume in Fig 2.
Fig 3 - please add a plot of DV, this would be interesting (even without a fitting line) - how many fetuses was it visible in? How many measurements could you obtain?
It would be interesting at this point to link the functional (Doppler) and structural measurements - did the authors find any links between these parameters and fetal/placental growth (e.g. UA vs fetal volume/placental volume (ie.. show does efficiency change? Does it predict growth/rate which would then be relevant to FGR))?
Table 2. Presumably the BPD measurements come from PM? This isn't described in the methods, probably brain volume/weight is also interesting and was estimated? BPD vs brain volume should be approx BV=(pi/6)*BPD^3
Can you generate a model of estimated fetal weight from HC/AC for the guinea pig (i.e. a hadlock style formula) and compare this with PM fetal weight? Same for placental?
Figure 4, Are there any numbers to go with this - this is a nice figure but it could be interesting to see the results for more than n=1.
Figure 5. units for BPD/weight are mm/g (these should be added), I'm unsure if this is the right way to present this given the dimension mismatch between length vs weight/volume.
Figure 6. These are nice plots and I think the basic fitting approach used previously is fine for this comparison. I'd really love to see a pair of plots for EFW and estimated placental weight if possible (but think the human data won't be there for this)
Discussion
Line 303/4 I think has an error and should link to the PM data only? Line 322: Doppler is misspelled.
Interestingly in the discussion the authors talk about RI rather than PI - what did the authors find with respect to RI in their study (it is easy to calculate)? Is it important to do this to make this comparison in the discussion?
Please add a section to describe the models that you fitted. They are data driven which is fine (apart from when the models contradict (as in Fig 1)), but there are plenty of physiological growth models that could have been applied (exponential/logarithmic). The discussion should trade off the data driven vs model-based approach both have advantages and limitations.
You could also comment on litter size being a relevant confounder here - the fetuses are not biologically independent so some evidence that they are statistically independent should be added.
Round 2
Reviewer 1 Report
The paper has been extensively revised with due attention to the referees' suggestions.